# Biosourced Multiphase Systems Based on Poly(Lactic Acid) and Polyamide 11 from Blends to Multi-Micro/Nanolayer Polymers Fabricated with Forced-Assembly Multilayer Coextrusion

**DOI:** 10.3390/ijms242316737

**Published:** 2023-11-24

**Authors:** Nour Jaouadi, Mohamed Jaziri, Abderrahim Maazouz, Khalid Lamnawar

**Affiliations:** 1CNRS, UMR 5223, Ingénierie des Matériaux Polymères, INSA Lyon, Université Claude Bernard Lyon 1, Université Jean Monnet, F-69621 Villeurbanne, France; nour.jaouadi@insa-lyon.fr (N.J.); khalid.lamnawar@insa-lyon.fr (K.L.); 2ENIS, Laboratoire Electrochimie et Environnement LEE, Université de Sfax, Sfax 3038, Tunisia; 3Laboratoire de Chimie Minérale Appliquée (LCMA) LR19ES02, Faculté des Sciences de Tunis (FST), Université Tunis El Manar, Tunis 2092, Tunisia; mohamedjaziri2003@yahoo.fr

**Keywords:** PLA, PA11, miscibility, compatibilization, coextrusion, rheology, morphology

## Abstract

The objective of the present study was to investigate multiphase systems based on polylactic acid (PLA) and polyamide 11 (PA11) from blends to multilayers. Firstly, PLA/PA11 blends compatibilized with a multifunctionalized epoxide, Joncryl, were obtained through reactive extrusion, and the thermal, morphological, rheological, and mechanical behaviors of these materials were investigated. The role of Joncryl as a compatibilizer for the PLA/PA11 system was demonstrated by the significant decrease in particle size and interfacial tension as well as by the tensile properties exhibiting a ductile behavior. Based on these findings, we were able to further clarify the effects of interdiffusion and diffuse interphase formation on the structure, rheology, and mechanics of compatible multilayered systems fabricated with forced-assembly multilayer coextrusion. The results presented herein aim to provide a deeper understanding of the interfacial properties, including the rheological, mechanical, and morphological behaviors, towards the control of the interface and confinement in multilayer polymers resulting from coextrusion, and also to permit their use in advanced applications.

## 1. Introduction

Among the novelties that have appeared on the thermoplastics market in recent years, a large part involves polymers derived from renewable resources, which currently constitute the strong point of innovation and development in the field of plastic materials [1,2,3]. Faced with the distrust in fossil resources and the resulting environmental problems, the polymers of natural origin constitute an alternative as they contribute effectively to reducing pollution caused by plastics and produce less greenhouse gases [4,5].

Given their lack of biodegradability, petroleum-based polymers are particularly unsuitable for short-term applications, which are the origin of pollution problems and the widespread dispersion of plastic microparticles in the environment. One of the strategies that could solve these difficulties would be to design materials that, once used, are easily eliminated/assimilated (degraded in vivo) by microorganisms in a natural environment [6]. According to their chemical nature, certain bio-resourced polymers have the advantage of also being biodegradable, which is essential in the field of packaging, the main consumer sector of plastics.

However, most of the currently existing biodegradable materials generally have limitations, making them far from satisfactory when it comes to competing with and replacing conventional petroleum-based polymers. Indeed, they require either chemical modifications or mixing with other polymers and additives in order to improve, for example, their thermal and mechanical properties or even to reduce their sensitivity to moisture. Thus, an objective is to be able to obtain mixtures with a controlled biodegradability and an adequate service life, exhibiting properties comparable to those of conventional polymers. 

Among the various families of bio-based polymers that have been developed, aliphatic biopolyesters occupy an important share of the market, and among them, polylactic acid or polylactide (PLA) plays a leading role [7,8]. This bioplastic constitutes an interesting alternative when it comes to replacing polymers derived from petrochemicals since it is produced from readily available renewable resources such as sugar beet or corn starch. Moreover, it is completely biodegradable, biocompostable, and biocompatible under certain conditions [9,10]. It is a thermoplastic polymer, either amorphous or semi-crystalline, already used in very various fields such as food packaging, biomedicine, agricultural films, and textile fibers. Nevertheless, in order for it to be used in applications with a high added value, its thermoplastic and barrier properties, its resistance to moisture, and its working conditions still need to be improved [11]. For this, the most used and studied method is mixing it with other bio-based polymers [12,13,14,15,16,17]. 

This last strategy has been widely studied for PLA, since polymer mixing is an effective method commonly used on an industrial scale, making it possible to combine the properties of different polymers in order to obtain a new material with a desirable behavior. Recently, polyamide 11 (PA11) has been considered a good candidate for blending with PLA, as it would help improve the barrier properties. Even if some studies have concluded a partial compatibility with high interfacial interactions between PLA and PA11 [18,19,20], the literature shows that the immiscibility and incompatibility of these polymers prevail, thus making compatibilization necessary to provide good interfacial interactions between the phases with the aim of generating super-resistant PLA-based systems. 

Various strategies have been studied, most of them targeting reactive pathways by adding a catalyst [21] to promote ester–amide interchange reactions, or a reactive copolymer containing glycidyl methacrylate (GMA). What makes such copolymers interesting is the fact that GMA is a chemical fragment with epoxide functions, and it can thus react with both hydroxyl and carboxyl reactive functional end chains of PLA and amine and carboxyl end groups of PA11. This coupling reaction at the interface improves the PLA/PA11 adhesion and makes the blend compatible. Walha et al. [22,23] have studied the incorporation of Joncryl^®^ in PLA/PA11 mixtures and the effects this had on the rheological, morphological, and mechanical properties of the materials. Joncryl^®^ is a multifunctional styrene-acrylic epoxy copolymer, generally used as a chain extender to improve the thermal stability of polyesters. Two main mixing methods that were used are as follows: the first involved introducing all the compounds simultaneously into the extruder and the second consisted of modifying the PLA by premixing it with Joncryl and then adding PA11. The authors demonstrated the role of Joncryl as a compatibilizer for the PLA/PA11 system, by showing the significant decrease in the size of the dispersed phases of PA11 and its interfacial tension as well as the improvement in its ductility, in particular with the second mixing pathway. Similar results were obtained by Zolali et al. [24], who used ethylene methyl acrylate-glycidyl methacrylate to compatibilize a co-continuous PLA/PA11 mixture. 

The first objective of the present study was to evaluate the properties of PLA/PA11 mixtures compatibilized with a multifunctionalized epoxide, Joncryl^®^, by reactive extrusion. Compared with the study of Walha et al. [22,23], our study was devoted to optimizing the mechanical and rheological behaviors of a PLA/PA11 80/20 wt% blend with 0.7 wt% of Joncryl using four different mixing strategies. Subsequently, the effect of these different strategies on the morphological, rheological, and thermomechanical properties as well as the structure of the PLA/PA11 blends was explored. 

The second objective was to evaluate the performance of the obtained blends processed by coextrusion to provide a better understanding of the interfacial behavior aiming towards the control of the interface/interphase and confinement in the multilayer polymer structures. Finally, a correlation between the properties obtained before and after coextrusion of the multilayer films was carried out for the optimal biosourced blend. 

## 2. Results

### 2.1. Morphological Properties of the Studied Uncompatibilized and Compatibilized Blends

The main objective of this section was to explore a PLA/PA11 blend system based on its morphological and interfacial properties. We studied the impact of incorporating a multifunctional epoxide (Joncryl) on the interfacial properties of PLA/PA11 blends prepared with melt extrusion using four different mixing strategies: (B1) the reactive in situ extrusion of all components at the same time; (B2) the modification of PLA with epoxide functions followed by the addition of PA11; (B3) the modification of PA11 by premixing it with Joncryl and then adding PLA, and (B4) premixing PLA and Joncryl followed by the addition of PA11 modified by Joncryl. The obtained blends were then compared with uncompatibilized PLA/PA11 mixtures (B0). 

SEM micrographs of PLA/PA11 blends processed according to the different strategies are presented in Figure 1. It can be clearly seen that the morphological properties were affected by the compositional changes in the mixture.

The SEM analyses illustrated a two-phase morphology with a nodular structure. In the case of the (80/20) uncompatibilized blend (Figure 1a), spherical particles of the minor PA11 phase were dispersed in the major PLA phase. The distribution of these particles was quite uniform, and their size distribution seemed rather narrow. Most of the droplets remained in the structure, while others were removed during the fracture, leaving empty cavities. This obtained morphology indicated the absence of miscibility between the two polymers PLA and PA11 as well as the presence of a high interfacial tension in the mixtures [22]. Self-compatibility, as presumed in the literature, seemed limited and insufficient.

For the compatibilized PLA/PA11 blends (B1: PLA/PA11/Jonc, obtained with one-step blending; B2: PLA_Jonc/PA11, obtained by modifying PLA by premixing it with Joncryl in an extruder and then adding PA11; B3: PLA/PA11_Jonc; obtained by modifying PA11 by premixing it with Joncryl and then adding PLA; and B4: PLA_Jonc/PA11_Jonc; obtained by premixing PLA and Joncryl followed by the addition of PA11 modified by Joncryl), the SEM micrographs presented in Figure 1b–e illustrate a significant reduction in the average size of the dispersed particles due to a lower interfacial tension [22]; a significant narrowing of the particle size distribution of the dispersed phase; a remarkable reduction in the voids at the interface and in the empty cavities corresponding to the extracted particles; and a better dispersion and more spherical particles of the minor phase. The increase in melt resistance and elasticity of PLA was suspected to influence the compatibilization properties. 

The SEM micrographs in Figure 1 show that B1 (b) and B2 (c) exhibited a good adhesion between their matrix and their dispersed phases. These results are evidenced by the total disappearance of the interstices at the interface and by the empty cavities corresponding to the extracted particles, indicating the decrease in the interfacial tension. Therefore, it was concluded that, by modifying the macromolecular chains of PLA with 0.7 wt% of Joncryl, the studied mixtures became much more compatible. 

The observed reduction in the diameter of the PA11 indicates that Joncryl had an emulsifying effect on the mixture, due to the suppression of coalescence. Such a phenomenon could be associated with a compatibilization due to the chemical reactivity of the epoxy function of Joncryl with the polymers of the mixture, i.e., the amine and acid chain ends of PA11 and PLA, respectively, as concluded by Walha et al. [22]. The morphological study confirmed that Joncryl acted as a compatibilizing agent. The challenge in the rest of the study was to understand the effect of this morphology on the thermal, rheological, and mechanical properties of the obtained mixtures.

### 2.2. Thermal Properties and Crystallinity

Table 1 summarizes the thermal characteristics of the PLA/PA11 blend (80/20) with and without multi-functionalized epoxide, while Figure 2 displays the thermograms of the second heating scan and the cooling cycle for the studied systems. 

The addition of PA11 to PLA slightly increased the degree of crystallinity of PLA (Xc) from 3% for pure PLA to 4.6% for the PLA/PA11 blend. This also influenced the cold crystallization temperature Tcc of PLA, which decreased to 106 °C for the uncompatibilized blend. The PA11 domains acted as nucleation centers, thus improving the crystallization of PLA in the mixtures [25]. 

The addition of Joncryl seemed to have a significant effect on the crystallization process of PLA; its presence induced an increase in the cold crystallization temperature Tcc of PLA, which changed from 108 °C for pure PLA to 120 °C for the modified PLA-Jcl. 

When it comes to the blends, it was clearly demonstrated that the studied mixtures remained biphasic systems as evidenced by the presence of two separate melting peaks corresponding to PLA and PA11, following the addition of PA11. However, a significant difference compared to the uncompatibilized mixture was observed.

For the PLA/PA11 blends, the presence of a single glass transition was observed. No significant changes concerning the glass transitions and the melting temperatures of each polymer in mixtures were seen.

For the compatibilized materials, the cold crystallization temperature Tcc of PLA increased, shifting from 106 °C (pure blend B0: PLA/PA11) to 115 for B1, 111 °C for B2, 110 for B3, and 112 °C for B4. This represents an increase of almost 9 °C compared to B1 and 5 °C compared to B2 for a total of 0.7 wt% of Joncryl. At the same time, a slight increase in ∆Hcc was observed, especially for B1 and B2, compared to the uncompatibilized mixture. 

For the different blends, it was also possible to note a second cold crystallization peak for PLA around 96 °C for B0 (PLA/PA11). The degree of crystallinity of the PLA in the blend decreased with the presence of Joncryl from 4.6% (pure blend B0: PLA/PA11) to 2.4% for the blend in which PLA was modified by Joncryl (B2: PLA-Joncryl/PA11). 

The changes in crystallization that were observed were related to the structural modification of the PLA chains induced by Joncryl. As a chain extender, Joncryl brought about an increase in the length of the chains and their molecular weight. Therefore, long-chain branched structures were formed in the Joncryl-modified PLA, thus increasing its molecular weight and decreasing the mobility of its chains. The presence of branches disturbed the settlement of the polymer chains, thereby preventing crystallization during the cooling step. As a result, the degree of crystallinity decreased. The reduced mobility of the chain was responsible for the increase in cold crystallization temperature [26,27,28]. This result confirmed the enhancement of the interfacial adhesion between PLA and PA11 due to reactions between the polymers and GMA functions [29]. 

During cooling, the presence of an exothermic peak was noted. It was attributed to the crystallization of PA11, and this crystallization temperature increased from 152 °C for PA11 to 161 °C for the modified PA11-Joncryl. For the compatibilization approaches, the appearance of a second exothermic peak was raised to 106 °C for B1 (PLA/PA11/Joncryl) and to 110 °C for B2 (PLA-Joncryl/PA11). This effect was associated with fractional crystallization that largely depended on the size distribution of the dispersed particles of the minority PA11 phase. As a result, the existence of nodules, of a significantly reduced size, generated an increase in the number of particles per unit volume and promoted fractional crystallization. This phenomenon appeared when the number of heterogeneities presented in the dispersed phase and that induced nucleation was less than the number of fine PA11 droplets generated in the medium. The behavior of the latter resembled that of germination and nucleation centers where the crystallization of PA11 emanated from a homogeneous germination mechanism. On the other hand, in the PA11 particles with a larger diameter, the nucleation phenomenon remained mainly heterogeneous with the presence of nucleons initiating crystallization. 

Several studies have been devoted to the phenomenon of fractional crystallization for a variety of polymer mixtures [30,31], and our results were in agreement with those obtained by Walha et al., 2017 [23], and Rasselet et al., 2019 [28], who studied PLA/PA11 blends with different compositions. They showed that, on the one hand, the addition of PA11 influenced the crystallization process of PLA by playing the role of nucleants, and, on the other hand, homogeneous germination took place within the PA11 particles when they were finely dispersed in the PLA matrix.

### 2.3. Shear Rheological Properties 

#### 2.3.1. Small Amplitude Oscillatory Shear (SAOS)

Figure 3 illustrates the evolution of the complex viscosity modulus η*, the storage modulus G′, and the loss modulus G″ versus the angular frequency of the neat PLA and PA11 as well as their counterparts modified with 0.7% of Joncryl at 195 °C.

It was clear that the incorporation of the multi-functionalized epoxide had a notable effect on the rheological behavior of both neat polymers PLA and PA11. Particularly for PLA, it increased the viscosity bringing it closer to that of PA11, especially at high frequencies. This can be explained by chain extension and/or a branching phenomenon [32,33,34,35,36,37]: the longer and heavier chains with short chain branches created more entanglements, which resulted in higher molecular weights exhibiting higher viscosities [23,32]. Also, the chain extension with 0.7% of Joncryl displayed a more pronounced shear thinning tendency of PLA, and consequently, it shifted the Newtonian plateau to lower angular frequencies [33]. The same trend was observed for PA11 with 0.7% of Joncryl. 

Moreover, we noted a significant increase in storage modulus G′ and loss modulus G″ of both PLA and PA11 with the addition of multi-functionalized epoxide especially at high angular frequencies (Figure 3b,c). This highlighted the improvement in the viscous and elastic behaviors of PLA and PA11. 

Figure 4 illustrates the complex viscosity modulus, the storage modulus, and the loss modulus versus the angular frequency at 195 °C for the B0 (80/20/0), B1 (80/20/0.7), B2 (80_0.7)/20, B3 (80/20_0.7), and B4 (80_0.35/20_0.35) PLA/PA11/Joncryl blends.

The curve of complex viscosity for the neat PLA/PA11 B0 blend was located between the values obtained for the neat polymers as shown in Figure 4a, thus diverging from the rule of mixture. B0 exhibited a higher shear-thinning behavior than pure PLA. Its rheological behavior seemed to be controlled by PA11 at lower frequencies. This result confirms the efficiency of PA11 in improving the melt viscosity and elasticity of PLA in the blend [34,36,37,38,39,40].

For the modified PLA-Joncryl/PA11 blend, the complex viscosity modulus and the storage modulus decreased compared to those of the PLA/PA11/Joncryl blend. This was expected as a result of the epoxide functions of Joncryl in this case reacting with the carboxyl and hydroxyl groups of PLA instead of with the carboxyl and amine groups of PA11. We could also note that η* and G′ of the modified PLA/PA11 blends were higher than the corresponding values of its uncompatibilized counterpart at a low angular frequency. 

The first and second approaches (B1 and B2) described above aimed to increase the viscosity and elasticity of PLA in order to tailor its viscoelastic properties with the aim of making them resemble those of PA11. It should be noted that this improvement in rheological properties of the PLA/PA11 blend was more pronounced with 0.7 wt% of Joncryl reacting directly with PLA (B1 and B2) instead of with PA11 (B3). For the case of 0.35 wt% in the blend B4, there was no significant change in the behavior of the PLA-PA11 blend, which was believed to be due to an insufficient compatibilization between the two polymers. This result suggested once again the occurrence of chemical reactions between the glycidyl methacrylate groups of Joncryl and the carboxyl (–COOH)/hydroxyl (–OH) end groups of PLA and carboxyl (–COOH)/amino end groups (–NH2) of PA11. In the case of the PLA/Joncryl reaction, the glycidyl esterification of carboxylic acid end groups preceded the hydroxyl end group etherification [32,35].

Figure 5a shows η″ as a function of η′, which are typical Cole–Cole plots. For PLA/PA11 mixtures, we observed that the addition of PA11 led to a significant increase in the relaxation time, indicating that the properties of the mixtures followed those of the dispersed phase. However, for the compatibilization approaches, especially for B1 and B2, we noticed that the addition of Joncryl produced a very long relaxation time. This result was in accordance with that of the viscoelastic behavior. 

In order to ensure that the Joncryl acted as a compatibilizer, a Han plot, representing the storage modulus (G′) versus the loss modulus (G″) as presented in Figure 5b, was created. For the compatibilization approaches, with the incorporation of the multi-functionalized epoxide, i.e., B1, B2, and B4, the curves of the blends displayed a linear correlation and an almost identical slope, indicating compatibility in comparison with the PLA/PA11 reference blend (B0). The results corroborated the SEM analyses presented previously.

#### 2.3.2. Elongation Properties

Figure 6 displays the elongation viscosities η_e_ obtained at 195 °C versus time for the neat PLA and PA11 polymers and the studied blends at various Hencky strain rates ranging from 0.1 to 20 s^−1^.

The main observation was that the transient viscosity curves of both polymers were well superposed and close to the linear viscoelastic envelope (LVE) (3η_0_^+^(t)) determined from the relaxation spectra in small amplitude oscillatory shear (SAOS) measurements. PLA and PA11 (Figure 6a,b) did not show any strain hardening at the investigated shear rates, which was in good agreement with the ordinary behavior of the linear polymers under stretching known as strain softening [36].

Figure 6e–i show the transient extensional viscosity of uncompatibilized and compatibilized 80/20_PLA/PA11 blends with 0.7% Joncryl as a function of time. For the PLA/PA11 blends, the elongation viscosity tended to follow Trouton’s rule. The observed strain softening behavior of the PLA and PA11 polymers and the linear viscoelastic envelopes of the compatibilized PLA/PA11 blends were higher than those of their non-compatibilized counterparts. 

For all compatibilization approaches (Figure 6f–i), an increase in the extensional viscosities was observed across the range of strain rate, and this could be attributed to the stronger melt strengthening of the multifunctional epoxide Joncryl. The presence of 0.7 wt% of Joncryl induced a significant variation in the rheological elongation behavior of both PLA and PA11 (Figure 6c,d). Indeed, both a significant increase in the linear viscoelastic envelope and an apparent increase in the elongational viscosities were observed in particular for PLA. It should also be pointed out that there appeared a hardening behavior called “strain hardening”, which was amplified with the speed of deformation. These results confirmed a Ferrule effect in the enhancement of the elongational viscosity through the elongations of chains, in particular of the PLA matrix. Thus, the long chain connections (LCB) that formed favored the generation of a network of inter-macromolecular entanglement nodes, with a density with which the system could better resist elongation. 

### 2.4. Mechanical Properties of the PLA/PA11 Blends 

The evolution of tensile properties (tensile modulus and elongation at break) as a function of the blend composition is shown in Table 2. It can be clearly observed that neat PLA had a high stiffness and brittleness (E ≈ 2084 MPa and εr ≈ 3%). The mechanical properties of the obtained PLA/PA11 blend changed drastically when PA11 (E ≈ 200 MPa and εr ≈ 225%) was incorporated into the mixture.

Table 2 shows that the incorporation of Joncryl slightly improved the tensile modulus of both PLA and PA11; however, the brittle properties of PLA were reduced by the addition of PA11, confirming the choice of this material to improve the mechanical properties of PLA. As a result, the elongation at break increased by adding PA11 (εr increased from 3% for pure PLA to 25% for the PLA/PA11 blends).

With the incorporation of Joncryl into the PLA/PA11 blends (Table 2, a significant improvement in mechanical properties was obtained, especially the elongation at break which increased from 25% for PLA/PA11 (80/20) to 255% for PLA/PA11/Joncryl B1 (80/20/0.7) and up to 300% for PLA_Joncryl/PA11 B2 (80_0.7/20). For PLA/PA11_Jcl B3 (80/20_0.7) and PLA_Jcl/PA11_Jcl B4 (80_3.5/20_3.5), it was 240% and 270%, respectively. 

For strategies B1 and B3, it is obvious that the addition of 0.7% by weight of Joncryl has little impact on the Young’s modulus (E), which remains practically constant around 1713 and 1690 MPa, while for the two other strategies B2 and B4, the addition of 0.7% by weight of Joncryl has an influence on the Young’s modulus (E), which changes from 1703 MPa for B0 to 1432 MPa and 1600 MPa, respectively, for B2 and B4.

These results indicate that a reactivity control took place at the interface between PLA, PA11, and GMA functions and also confirm the positive effect of the multifunctional epoxide chain extender on the compatibilization between PLA and PA11. The obtained mechanical properties corroborated the results from the SEM microscopy and rheological investigations.

### 2.5. Multilayer Coextrusion 

#### 2.5.1. Sample Preparation: Processing of Multilayers

According to the morphology, the rheological, and mechanical results of the mixtures in the first part, only two compatibilization methods (B1 and B2) will be used with the blend of PLA/PA11 (B0) to prepare multilayered systems fabricated with forced-assembly multilayer coextrusion. 

All investigated multilayer films are listed in Table 3, where n is the number of multipliers, and N is the corresponding number of layers. The estimated nominal layer thickness for each layer with a C/A/C film configuration was calculated using Equation (1):(1)$$h_{\mathrm{nomA},C}={Ø}_{A,C}\;\frac{h_{\mathrm{total}}}{2^{n}},$$
and here, Ø_A_ is the volume fraction of polymer A in the film estimated with the weight compositions and densities at the extrusion temperature, 2^n^ is the number of A layers with n being the number of multipliers, and h_total_ is the total thickness of the multilayer film. This equation worked similarly for both A and C components with different volume fractions (20% for C and 80% for A). The compositions of the studied blends for coextrusion are provided in Table 4. 

#### 2.5.2. Rheological Study of Multi-Micro/Nanolayered Polymers and Blends (Linear Viscosity)

Dynamic frequency sweep tests were performed under a fixed strain amplitude of 3%, which was in the linear viscoelastic regime, and an angular frequency of 628 to 0.05 rad/s, at a temperature of 195 °C. Figure 7 depicts the dynamic complex viscosity (η*) against the angular frequency (w) for the neat polymers, their 80/20 compatibilized mix (B1 and B2), the uncompatibilized blend (B0), and the multilayer structures of the systems at 195 °C. 

As it can be observed, the viscosity of the multilayer structures decreased when the number of layers increased from 3 to 1535 for the neat polymers PLA and PA11 and the blends. This was due to the lack of entanglements in the interfacial region, leading to the appearance of interfacial slip at their interface. The negative deviation of viscosity in the multilayers was believed to be caused when the increase in the number of layers greatly enhanced the interfacial contribution, and then reduced entanglements between polymer chains (results of the SEM analyses are shown here for the sake of clarity).

The complex viscosity of the blends is shown in Figure 7c–e for the neat and multilayer structures from 3 L to 1535 L. The multilayer structures presented a lower viscosity than the blends of the neat materials before coextrusion, suggesting a great negative deviation and pronounced interfacial slippage. It was seen that more the number of layers, negative deviation can be observed at low frequencies, thus explaining the immiscibility between the PLA/PA11 layers, and PA11 truly confines PLA during coextrusion.

#### 2.5.3. Architecture and Morphology of Coextruded Multi-Micro/Nanolayers

Figure 8 shows the SEM micrographs of the multilayer blend layers B0, B1, and B2 from the microscale to the nanoscale ranging from 3 to 1535 layers with an identical total thickness of 250 µm, in order to, on the one hand, study the effect of increasing the number of layers on the morphologies of the multilayer films and, on the other hand, evaluate the dispersion and the distribution of the minority component in the continuous matrix and estimate the adhesion at the interface of the two phases. In the C/A/C configuration, PA11 was used to confine the reference blend B0 and the blends compatibilized by Joncryl, i.e., B1 and B2. It was also an external layer that protected the mixtures.

As shown in the figure, uniform and continuous layers were present, with sharp interfaces and a continuous structure for the films with 3 to 1535 layers. In the SEM micrographs for the micro-layered 47 L and 1535 L films, all the B2 and PA11 layers were clearly distinguished and continuous, and the thickness of the layers was noticeably irregular for a composition of 80/20 where PA11 was used to confine the PLA layers. When the number of layers increased further, they could no longer be distinguished.

When increasing the number of layers from 47 to 1535 (Figure 9), from the micrometric scale to the nanometric scale, the nominal thickness of each layer decreased, which was seen as a reduction in size of the spherical particles of the minority phase PA11. This confirmed the effect of confinement by PA11 on the structural and morphological properties in the multilayered systems fabricated with forced-assembly multilayer coextrusion.

#### 2.5.4. Study of the Mechanical Properties of Multilayer Systems 

The impact of increasing the number of layers and the composition on the mechanical properties is shown in Table 5. A significant improvement in mechanical properties was obtained when transitioning from 3 to 1535 layers as compared to the mechanical properties of the blends before the coextrusion process. The increase in the number of layers implied a reduction in their nominal thickness, thus improving the flexibility of the films by lowering the tensile modulus in the elastic range. For instance, the PLA/PA11 blend went from 1703 MPa to 1640 MPa, which was somewhere in between the neat PLA and neat PA11, as shown in Table 5. For blend B2, it dropped from 1432 MPa to 1328 MPa, and we obtained an increase in the elongation at break, in particular for B2, from 300% before coextrusion to 355% for 1535 layers with a nominal layer thickness of less than 100 nm. This may be explained by the impact of geometric confinement of the PA11 crystals against the amorphous PLA during the coextrusion process. 

## 3. Discussion

Throughout this study, we have demonstrated that PLA and PA11 are immiscible polymers, as judged from their observed two-phase morphology and their nonlinear correlation seen in the Han plot. To improve the homogeneity of the system, we used four compatibilization methods using reactive extrusion: (B1) one-step mixing of the three components PLA, PA11, and Joncryl; (B2) premixing of PLA and Joncryl followed by the addition of PA11, (B3) premixing of PA11 and Joncryl followed by the addition of PLA, and (B4) premixing of PLA and Joncryl followed by the addition of PA11 modified by Joncryl. Studies of the morphological, thermal, rheological, and mechanical properties of the blends revealed that Joncryl acted as a compatibilizer. The goal was to study the impact of these different approaches on the structure and characterizations of the blends in order to determine the composition best suited for the production of multilayer films intended for food packaging. Regarding the results obtained with the base PLA/PA11 blend, morphological analysis conducted using SEM revealed minor adhesion at the interface between the two polymers, especially for the 80/20 composition blend in which PA11 constitutes the dispersed minority phase. DSC thermal analysis showed a decrease in the cold crystallization temperature of PLA in the blend. Rheological properties studied in dynamic melt state showed that the shear-thinning behavior of PLA/PA11 blends is more pronounced than that of pure PLA. The presence of PA11 in the blend appears to govern the rheological behavior, especially at low frequencies. Tensile mechanical properties of the prepared blends indicated that the presence of PA11 led to a loss of stiffness and tensile strength but, in return, induced a significant increase in ductility. The behavior of PLA/PA11 blends is evidently dominated by the major matrix component.

Moving on to the compatibilized blends, in the first approach (B1), where all three components were introduced simultaneously into the extruder, this led to the in situ formation of a PA11-Joncryl-PLA copolymer. This copolymer acts as a coupling agent, promoting both the adhesion between the blend’s components and the reduction of interfacial tension. The melt-state rheological properties showed a significant increase in viscosity and storage modulus. Thermal analyses revealed a reduced ability of PLA chains to crystallize due to the formation of entanglement networks between PLA chains and the formed copolymer. Mechanical properties were also improved, with a notable increase in ductility while maintaining good stiffness.

The second approach (B2) resulted in the in situ formation of a PLA-Joncryl-PLA copolymer, which, based on scanning electron microscopy (SEM) fracture surface observations, also promotes the adhesion between the PA11 and PLA phases of the blend. This copolymer appears to be perfectly miscible in the system and causes the plasticization of the PLA component. This significantly increased the extensibility of the matrix chains, thus enhancing its flexibility and ductility.

The third approach (B3) led to the in situ formation of a PA11-Joncryl-PA11 copolymer, which, according to SEM cryofracture observations, maintains a nodular biphase morphology. Melt-state rheological properties showed a slight increase in viscosity and storage modulus that remained close to those of PLA. Thermal analyses revealed a reduction in the ability of PLA chains to crystallize due to the formation of entanglement networks between PLA chains and the formed copolymer. However, this was insufficient to significantly improve the compatibility of the two polymers (PLA and PA11). Mechanical properties were also improved, with a significant increase in ductility, while maintaining good stiffness.

The fourth approach (B4) involved introducing Joncryl into both the PLA and PA11 components while maintaining a constant total Joncryl content of 0.7% in the blend. SEM fracture surface observations showed that the nodular biphase morphology was well preserved. Melt-state rheological properties indicated a significant increase in viscosity and storage modulus. Thermal analyses revealed a decrease in the ability of PLA chains to crystallize. This was attributed to the formation of entanglement networks between PLA chains and the formed copolymer. Adhesion remained insufficient to significantly improve the compatibility of the two polymers, PLA and PA11.

Based on the results of morphological, rheological, and mechanical tests of the mixtures, only two compatibilization methods (B1 and B2) in addition to the blend of neat PLA/PA11 (B0) were subsequently used to prepare multilayered systems obtained with forced-assembly multilayer coextrusion of 3 to 1535 layers. It was found that, by increasing the number of layers, the nominal thickness of each layer was reduced, which led to a smaller size of the spherical particles of the minority phase PA11. This in turn confirmed the confinement effect of PA11 on the structural and morphological properties in the multilayered systems. Moreover, a significant improvement in the mechanical properties was obtained when increasing the number of layers, thus confirming the role of confinement on the rheology and morphology of the multilayer polymeric structures. It was also seen that the viscosity of the multilayer structures decreased as the number of layers increased. A negative deviation was observed at low frequencies, which explains the immiscibility between the PLA/PA11 layers and why PA11 truly confines PLA during the coextrusion process.

## 4. Materials and Methods

### 4.1. Materials

The PLA (grade: 4032D) used in this study was in the form of pellets and was purchased from NatureWorks (Minnetonka, MN, USA). It had a D-isomer content of approximately 4%, with an average molecular weight of 100,000 g/mol (according to GPC analysis), and a glass transition temperature and a melting temperature of approximately 59 °C and 168 °C (according to DSC analysis), respectively. PA 11 (grade: BESNO P40 TL) was supplied by Arkema (Colombes, France) under the trade name Rilsan. A commercially modified acrylic copolymer with epoxy functions, Joncryl ADR^®^-4368, accepted by the Food and Drug Administration (FDA) for food packaging, was obtained from BASF (Ludwigshafen, Germany). Its average functionality on epoxide was 9. The specific chemical structures and the main characteristics of the employed materials are reported in Figure 10 and Table 6.

### 4.2. Blend Preparation Methods

#### 4.2.1. Reactive Extrusion

Before compounding, pellets of both PLA and PA11 were dried under vacuum at 80 °C for 12 h to remove moisture, while Joncryl ADR was dried under vacuum at 40 °C. Blends were then prepared in a corotating twin-screw extruder (Thermo Electron Polylab System Rhecord RC400P, Courtaboeuf, France) with a screw diameter of 16 mm and an L/D ration of 25:1. The screw rotation speed was 40 rpm, the total residence time was set to 3 min, and the temperature profile was fixed at 145 °C (feeding zone), 195, 200, 195 °C (melting zones), and 195 °C (die). The twin-screw extruder was used for (i) the processing of the neat polymers at different temperatures, shear rates, and residence times, (ii) reactive extrusion to investigate the role of Joncryl ADR^®^ in the control of the degradation of PLA and PA11, and (iii) the elaboration of uncompatibilized and compatibilized PLA/PA11 blends. After melt blending, each extrudate was quenched in a cold water bath and granulated. 

Four mixing strategies were carried out (Figure 11): (a) one-step mixing of all components, PLA, PA11, and Joncryl; (b) premixing of PLA and Joncryl followed by the addition of PA11, (c) premixing of PA11 and Joncryl followed by the addition of PLA, and (d) premixing of PLA and Joncryl followed by the addition of PA11 modified by Joncryl. 

For the first approach, PLA, PA11, and Joncryl were mixed simultaneously in the twin-screw extruder. For the other approaches, the blends were prepared in two steps. For the second approach, PLA and 0.7 wt% of Joncryl were first mixed together in the twin-screw extruder, quenched in cold water, and granulated. Then, modified PLA pellets (named PLA-J0.7) were dried under vacuum at 60 °C for 12 h, and subsequently mixed using twin-screw extrusion with neat PA11. The compositions of the studied blends are provided in Table 7. An amount of 0.7 wt% of Joncryl was used for PLA, PA11, and the PLA/PA11 (80/20) blends.

#### 4.2.2. Compression Molding

Most of the samples for shear rheology measurements to study linear viscoelasticity were prepared using hot compression. Before processing, all the polymers were first dried in a vacuum oven for at least 48 h to remove any moisture. The dried polymers were then compression-molded into disks and/or rectangular sheets between two Teflon films to obtain a smooth surface at a temperature of 190 °C and a pressure of 200 bar.

#### 4.2.3. Multilayer Coextrusion

Film structures were fabricated with forced-assembly multilayer coextrusion using a two-component coextruder equipped with layer multiplying elements (LME) (multipliers) as schematically shown in Figure 12. This coextrusion system was composed of two single-screw extruders (A and C), a coextrusion feed block, a set of vertical layer multipliers, a flat exit die, and a thermally regulated chill roll. From the feed block where the two melt streams combined, the initial three-layer flow went through a series of multipliers. The melt was initially sliced vertically, and then, the halves were spread horizontally to the original width and finally recombined, while keeping the total thickness of the melt constant, thus doubling the number of layers and reducing the thickness of each layer by a factor of two after each multiplier. The number of layers (N) in the as-coextruded films was determined by the number of multiplier elements (n) as N = (3 × 2^n^) − 1, (n ˃ 0) with C/A/C configuration systems through the feed block, which made it possible to have an initial number of three layers by multiplying the multiplier elements, thus leading to the fabrication of multilayered films. 

A calibration curve of flow rate versus screw speed for each polymer was created beforehand by adjusting the screw speed of extruders between 10 and 69 rpm. The thickness ratio of the layers could be changed by varying the flow rate of each polymer melt. In this study, multilayer films with 3, 47, 383, and 1535 layers with an identical total thickness of 250 µm were investigated. The nomenclature for the multilayer sample was “NL”, where N is the number of layers. 

### 4.3. Experimental Methods and Procedures

#### 4.3.1. Differential Scanning Calorimetry (DSC) 

A differential scanning calorimetry (DSC) (Q23) from TA Instruments (New Castle, DE, USA) was used to measure the thermal characteristics of the blends. Samples of around 5 mg were cut from the pellets and added to sealed aluminum pans. All the experiments were performed under dry nitrogen as a protective gas (50 mL/min). Three calorimetric scans were carried out for each sample at a heating or cooling rate of 10 °C/min. The first heating scan, in which the thermal history was suppressed, was performed from 30 °C to 220 °C before a 3 min isothermal scan at 220 °C was applied. Then, the cooling scan went from 220 °C to 30 °C, followed by another 3 min isothermal scan at 30 °C, and finally, the second heating scan was performed from 30 °C to 220 °C. 

The glass transition temperature (Tg), cold crystallization temperature (Tcc), enthalpy of cold crystallization (ΔH_cc_), melting temperature (Tm), and melting enthalpy (ΔH_m_) were recorded from the second heating cycle. The degree of crystallinity (X_C_) due to PLA and PA11 in a blend was calculated using Equation (2):(2)$$X_{c}\left(\%\right)=100*\frac{\Delta H_{m}-\Delta H_{cc}}{f*\Delta H_{m\infty }},$$
and here, ΔH_m_ is the heat of fusion of the component, ΔHcc is the cold crystallization enthalpy (J/g); ΔH_m∞_ corresponds to the heat of fusion for 100% crystalline polymer, and “f” is the weight fraction of PLA or PA11 in the blends. We considered ΔH_m∞_ (PLA) = 93 J/g and ΔH_m∞_ (PA11) = 200 J/g [38,39]. 

#### 4.3.2. Scanning Electron Microscopy (SEM) 

Scanning electron microscopy was used to describe the morphology of the PLA/PA11 blends. The structures were examined using a Hitachi S3500 microscope (Tokyo, Japan) under an accelerating voltage of 5 kV. Before examining the surfaces, they were cryogenically fractured in liquid nitrogen and covered with gold–palladium to avoid an electrostatic charge for the duration of the analysis. 

#### 4.3.3. Rheological Properties 

Small amplitude oscillatory shear measurements (SAOS)

The measurements were executed in a DHR-2 stress-controlled rotational rheometer from TA instruments using a plane–plane configuration, a plate diameter of 25 mm, and a gap inferior to 2 mm. The temperature was set to 195 °C and a nitrogen flux was used to avoid thermal degradation of the neat polymers and their blends. The linear domain was put into effect under a stress of 0.1 rad/s, which expanded from 0.2 to 1000 Pa. Dynamic frequency sweep tests were performed at 20 Pa over an angular frequency range of 0.05 to 628 rad/s. 

Elongation rheological properties

Uniaxial extensional experiments were conducted using a second-generation Sentmanat Extensional Rheometer Universal Testing Platform (SER-G2) coupled to the stress-controlled rotational rheometer (DHR2) from TA instruments. With this fixture, a constant Hencky strain rate was imposed by applying a specific angular velocity to turn the dual wind-up drums. Strain validation was performed during the start-up measurements by the virtue of the camera installed in the oven of the rheometer. Sheet samples for extensional measurements were prepared by compression molding at 195 °C, after which rectangular plates were cut in the following dimensions: a thickness of 0.5 mm, a width of 10 mm, and a length of 20 mm. The specimens were clamped between the two counter-rotating wind-up drums. Before starting the experiments, the test chamber was heated to 195 °C and kept under a nitrogen atmosphere to avoid the degradation of the samples. The specimens were elongated in the melt up under very large strains (reaching 3.8 of the Hencky strain). Extensional measurements were performed at constant strain rates from 0.1 to 20 s^−1^. 

The material function, usually determined with extensional rheometry, is the tensile stress growth coefficient (extensional viscosity) defined from the tensile stress (σ_E_) by the following:(3)$${ɳ}_{E}^{+}\left(t,ἐ\right)=\frac{\sigma_{E}\left(t,ἐ\right)}{ἐ}$$

At sufficiently small strain rates, the extensional viscosity can be derived from the Boltzmann superposition principle as follows [22,23]: (4)$$\underset{ἐ\to 0}{\lim}{ɳ}_{E}^{+}(t,\;ἐ)=\;{ἐ}_{E}^{+}\left(t)=3\int_{0}^{t}G\left(s\right)ds=3{ɳ}^{+}(t\right)$$

Here, G(s) is the stress relaxation modulus, and ɳ^+^(t) is the shear stress growth coefficient obtained from start-up shear (at a fixed shear rate).

In the limit of the vanishing strain rate (i.e., zero-rate limit), the extensional viscosity simply becomes three times the zero-shear viscosity (ɳ_0_):(5)$$\underset{ἐ\to 0}{\lim}{ɳ}_{E}^{+}(ἐ)=3\int_{0}^{\infty }G\left(t\right)dt=3{ɳ}_{0}$$

This is referred to as the linear viscoelastic (LVE) envelope, which is always used to normalize extensional viscosity data.

#### 4.3.4. Tensile Properties

Mechanical properties were determined using an Instron machine according to the ASTM method D638 under ambient temperature (25 °C) with a cross-head speed of 5 mm/min. The dimensions of the dumbbell-shaped specimen were 25 mm in length, 5 mm in width, and 2 mm in thickness.

## Figures and Tables

**Figure 1 ijms-24-16737-f001:**
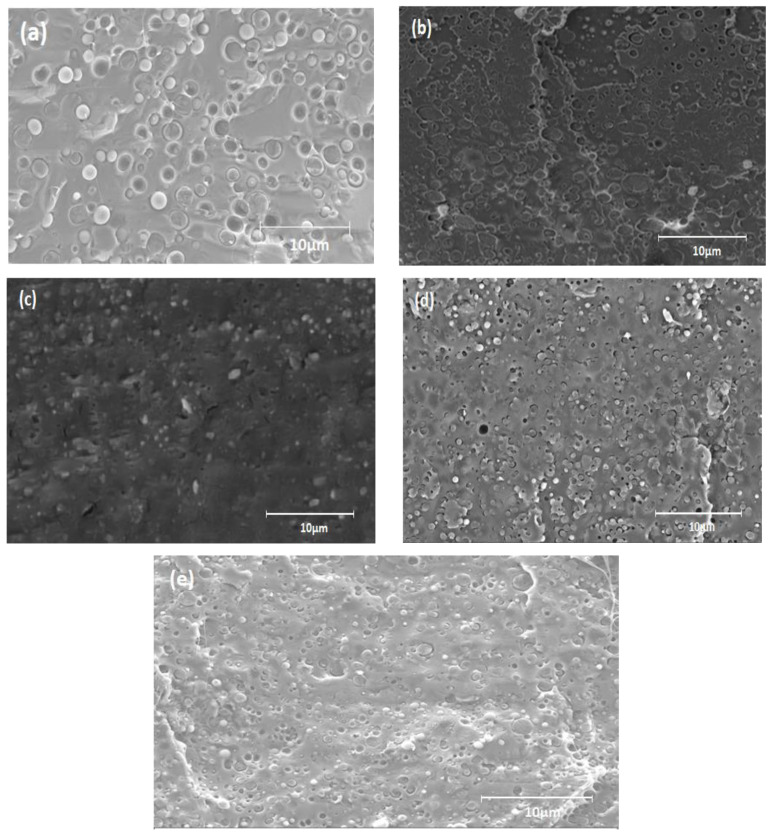
SEM micrographs of uncompatibilized and compatibilized PLA/PA11 blends: (**a**) (B0: 80/20), (**b**) (B1: 80/20/0.7), (**c**) (B2: 80_0.7/20), (**d**) (B3: 80/20_0.7), and (**e**) (B4: 80_0.35/20_0.35).

**Figure 2 ijms-24-16737-f002:**
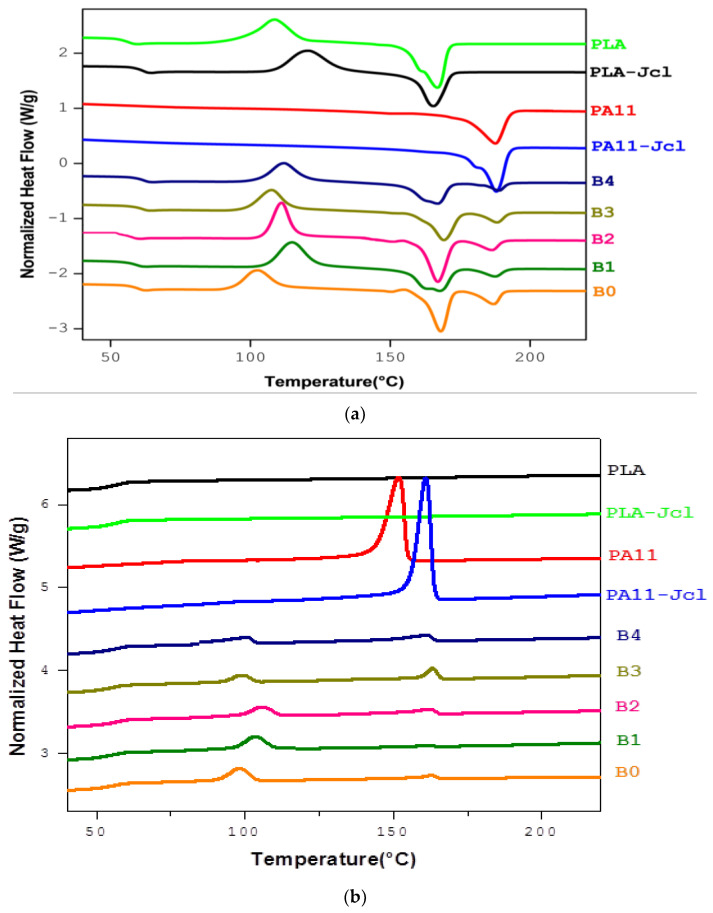
DSC thermograms of (**a**) the second heating cycle and (**b**) the cooling cycle of the neat polymers and the blends.

**Figure 3 ijms-24-16737-f003:**
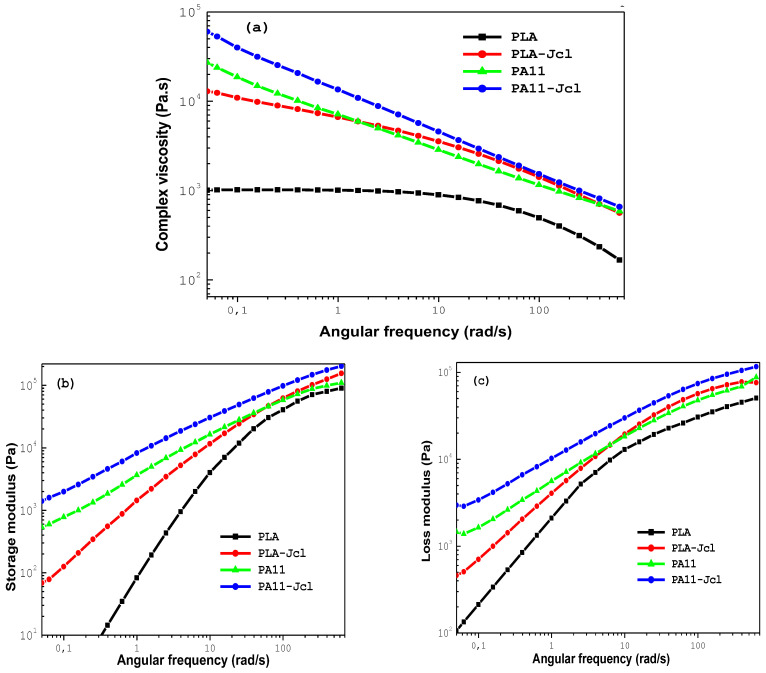
Variation of the complex viscosity modulus η* (**a**), the storage modulus G′ (**b**), and the loss modulus G″ (**c**) versus the angular frequency of neat PLA and PA11 and their counterparts modified with 0.7% of Joncryl at 195 °C.

**Figure 4 ijms-24-16737-f004:**
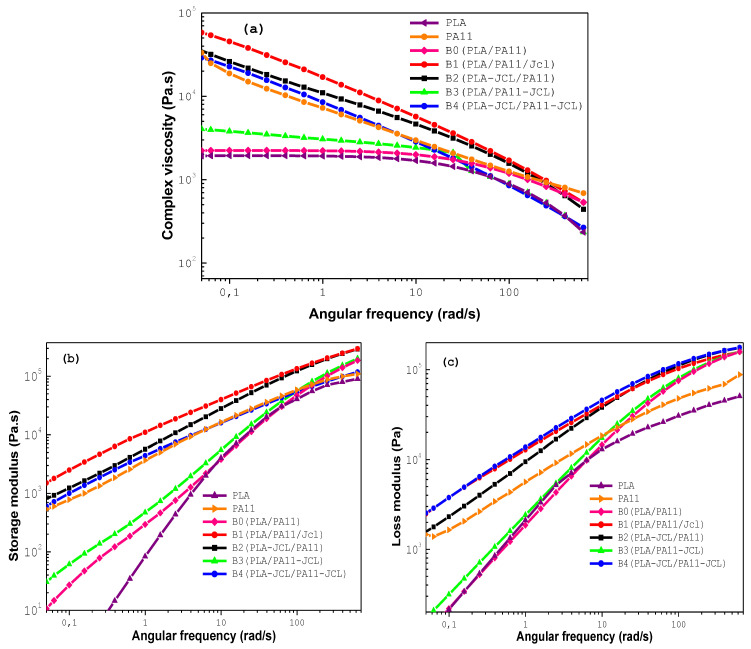
(**a**) The complex viscosity modulus, (**b**) storage modulus, and (**c**) loss modulus versus the angular frequency at 195 °C for the various compatibilized blends.

**Figure 5 ijms-24-16737-f005:**
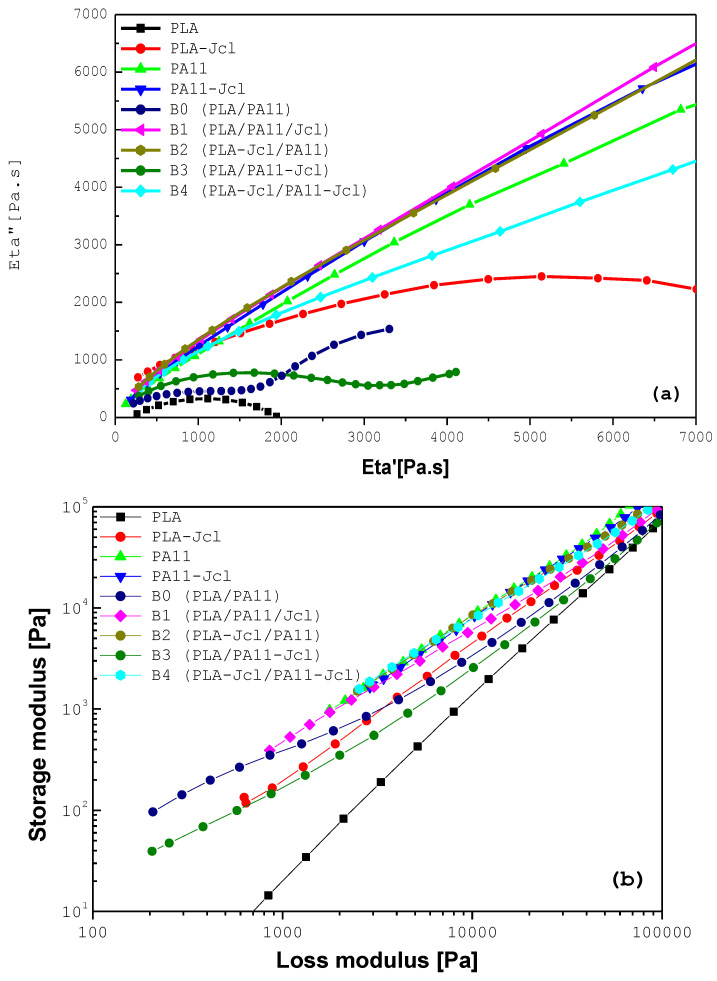
(**a**) Cole–Cole plots at 195 °C for PLA, PA11, their modified counterparts with 0.7% of Joncryl, the PLA/PA11 blends, and the compatibilized blends. (**b**) Han plot for neat and modified PLA and PA11 as well as their uncompatibilized and compatibilized 80/20 blends with 0.7% of Joncryl.

**Figure 6 ijms-24-16737-f006:**
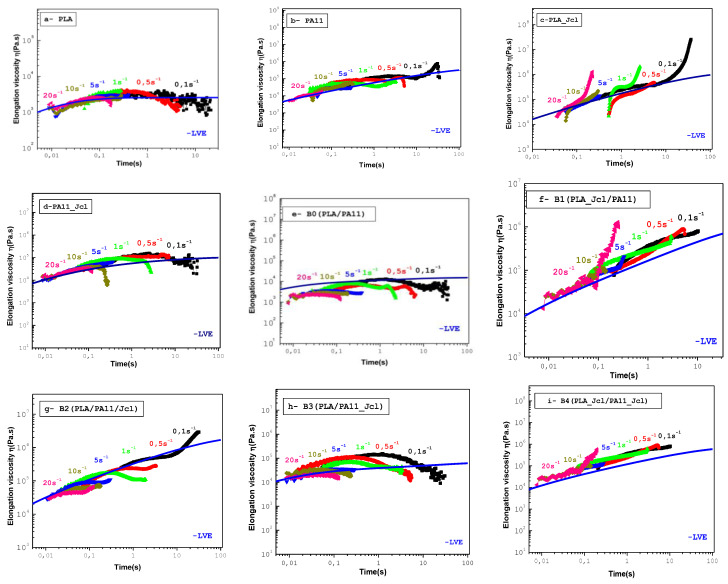
Extensional viscosity at a temperature of 195 °C with the extensional rate varying from 0.1 to 20 s^−1^ for neat and modified PLA and PA11 polymers (**a**–**d**) and studied blends B0 (**e**), B1 (**f**), B2 (**g**), B3 (**h**), and B4 (**i**). (The solid green line represents the LVE (3η_0_^+^(t)), determined from the relaxation spectra (3η_0_^+^(t) = 3 ∫ G(s)ds).)

**Figure 7 ijms-24-16737-f007:**
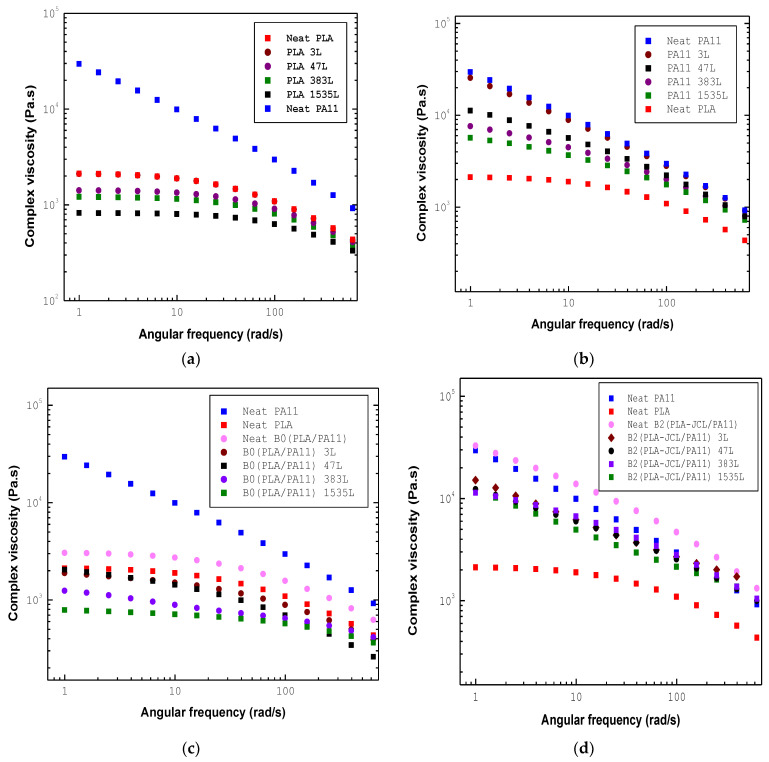
Complex viscosity versus angular frequency for (**a**) neat PLA and (**b**) neat PA11 and (**c**) B0 (PLA/PA11), (**d**) B1 (PLA/PA11/JCL), and (**e**) B2 (PLA-JCL/PA11) multilayer films from 3 L to 1535 L at 195 °C.

**Figure 8 ijms-24-16737-f008:**
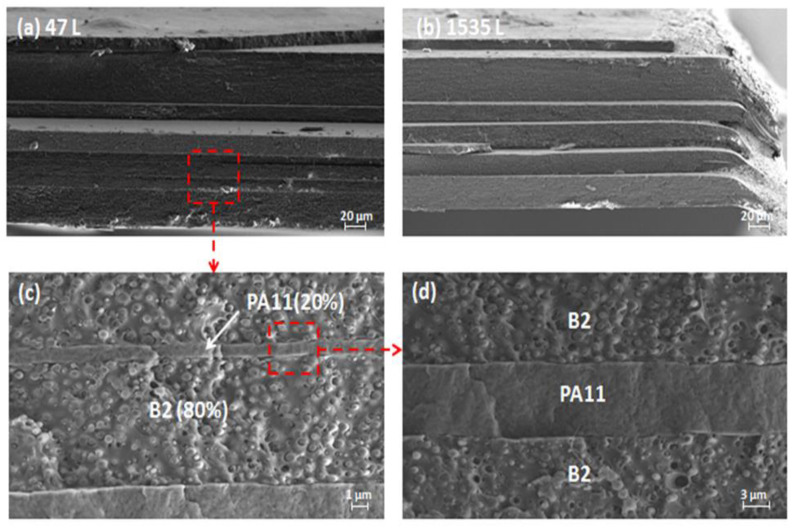
SEM micrographs of (**a**) 47 L and (**b**) 1535 L in a C/A/C configuration for PA11/B2/PA11 multilayered structures. (**c**,**d**) show respectively a zoom of B2/PA11 layers in 47 L systems with 1 µm and 3 µm of scales.

**Figure 9 ijms-24-16737-f009:**
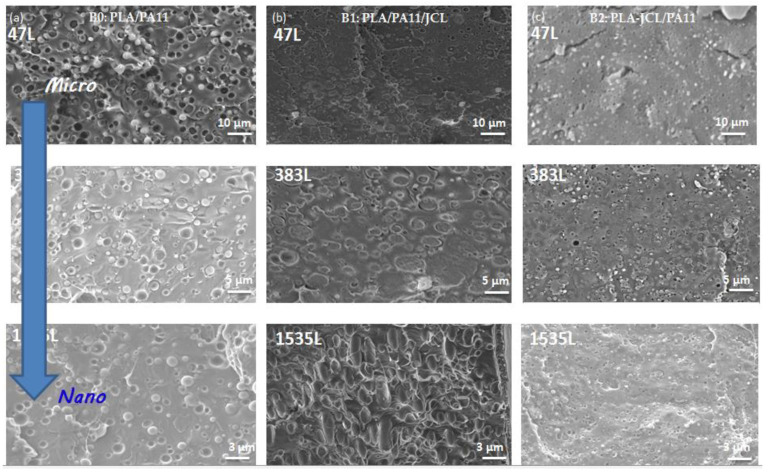
SEM micrographs of the multilayer systems, i.e., (**a**) B0, (**b**) B1, and (**c**) B2, with a number of layers ranging from 47 L to 1535 L.

**Figure 10 ijms-24-16737-f010:**
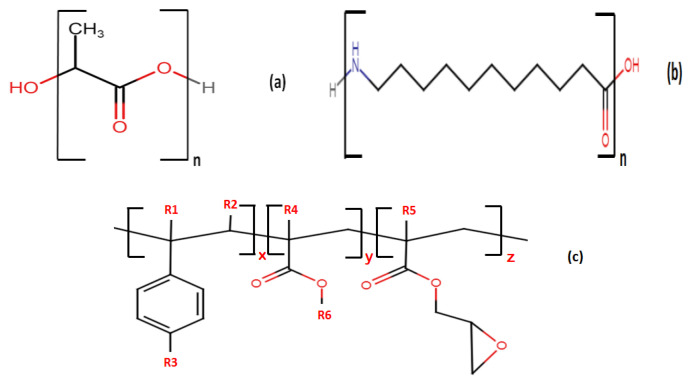
Chemical structure of (**a**) PLA, (**b**) PA11, and (**c**) Joncryl ADR^®^-4368, and the general structure of the styrene-acrylic multifunctional oligomeric chain extenders. R1–R5 are H, CH_3_, a higher alkyl group, or combinations of them; R6 is an alkyl group. X, Y, and Z are between 1 and 20 [37].

**Figure 11 ijms-24-16737-f011:**
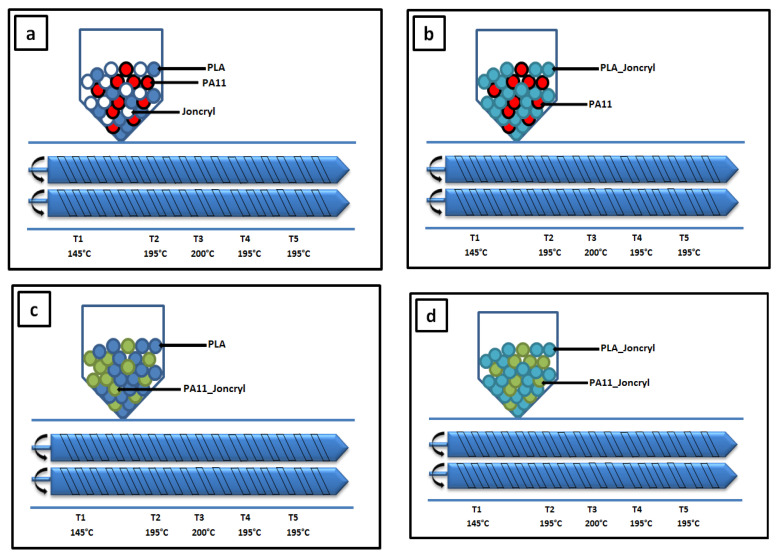
The four mixing strategies using a corotating twin-screw extruder with a 16 mm diameter: (**a**) B1, one-step mixing of PLA, PA11, and Joncryl; (**b**) B2, mixing PA11 with modified PLA; (**c**) B3, mixing PLA with modified PA11; and (**d**) B4, mixing PA11_Joncryl with PLA_Joncryl. (**e**) The synopsis of the experiment.

**Figure 12 ijms-24-16737-f012:**
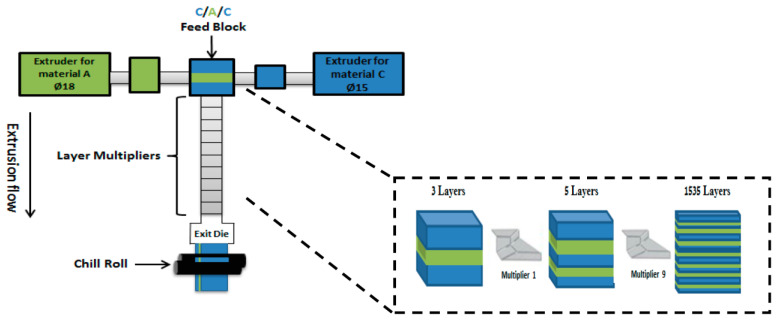
A schematic illustration showing the layer multiplication in the two-component (A and C) coextrusion process equipped with the multiplying-element device for the C/A/C configuration.

**Table 1 ijms-24-16737-t001:** Thermal properties of the neat polymers and the blends.

Samples	Tg (°C)		Tcc (°C) PLA	∆Hcc (J/g) PLA	Tc (°C) PA11	Tcc2 (°C) PLA	Tm (°C)		∆Hm (J/g)		Xc (%)	
	PLA	PA11					PLA	PA11	PLA	PA11	PLA	PA11
PLA	59	-	108	35.3	-	-	168	-	38	-	3	-
PLA-Jcl	61	-	120	36.8	-	-	166	-	39	-	2.4	-
PA11	-	-	-	-	152	-	-	188	-	45	-	22.5
PA11-Jcl	-	-	-	-	161	-	-	188	-	41	-	20.6
B0 PLA/PA11	60		106	22.2	163	96	169	189	25.6	7.4	4.6	18.5
B1 PLA/PA11/Jcrl	58		115	30.1	161	106	168	188	32	5.6	2.6	14.1
B2 PLA-Jcl/PA11	57		111	29	162	110	167	186	30.8	7.1	2.4	17.8
B3 PLA/PA11-Jcl	60		110	23.8	163	99	169	188	26.1	6.3	3.1	16.3
B4 PLA-Jcl/PA11-Jcl	61		112	24.2	164	102	167	189	26.4	4.9	3	12.5

**Table 2 ijms-24-16737-t002:** Effect of Joncryl on mechanical properties of the studied blends.

Samples Name	Compositions	Tensile Modulus (MPa)	Elongation at Break (%)
PLA/PA11	100/0 0/100	2084 ± 45 200 ± 10	3 ± 0.5 225 ± 26
B0 (PLA/PA11)	80/20	1703 ± 50	25 ± 5
PLA_Jcl PA11_Jcl B1 (PLA/PA11/Jcl)	99.3/0/0.7 0/99.3/0.7 80/20/0.7	3010 ± 36 217 ± 5 1713 ± 10	2.8 ± 1 198 ± 13 255 ± 15
B2 (PLA_Jcl/PA11)	80_0.7/20	1432 ± 25	300 ± 20
B3 (PLA/PA11_Jcl)	80/20_0.7	1690 ± 30	240 ± 4
B4 (PLA_Jcl/PA11_Jcl)	80_3.5/20_3.5	1600 ± 26	270 ± 30

**Table 3 ijms-24-16737-t003:** Characteristics of the investigated multilayered PLA/PA11 films.

Feed Block	Polymer Systems (wt%)		PLA/PA11		
	No. of Layers (N)	No. of Multipliers (n)	* h_T_ (µm)	* h_N_	
				A	C
C/A/C	3 L	0	250	200 µm	25 µm
	47 L	4	250	12.5 µm	1.56 µm
	383 L	7	250	1.56 µm	195 nm
	1535 L	9	250	390 nm	49 nm

* h_T_: the total film thickness. * h_N_: nominal layer thickness.

**Table 4 ijms-24-16737-t004:** Composition and designation of the unmodified and modified PLA, PA11, and their blends.

Extruder	A (Ø18)	C (Ø15)
Material designation/composition	PLA	PLA
	PA11	PA11
80/20	B0 (PLA/PA11)	B0 (PLA/PA11)
	B1 (PLA/PA11/Jonc)	B1 (PLA/PA11/Jonc)
	B2 (PLA-Jcl/PA11)	B2 (PLA-Jcl/PA11)
	B0 (PLA/PA11)	PA11
	B1 (PLA/PA11/Jonc)	PA11
	B2 (PLA-Jcl/PA11)	PA11

**Table 5 ijms-24-16737-t005:** Mechanical properties of multilayers blends studied 1535 L.

Sample	Tensile Modulus (MPa)	Elongation at Break (%)
PLA-100	2060 ± 25	6 ± 0.5
B0 PLA/PA11-80/20	1640 ± 30	20 ± 4
PA11-100	175 ± 10	235 ± 14
B1 PLA/PA11/Jonc 80/20/0.7	1690 ± 32	260 ± 2
B2 PLA-Jonc/PA11 80_0.7/20	1328 ± 15	355 ± 20
PA11/B0/PA11	1556 ± 25	80 ± 30
PA11/B1/PA11	1320 ± 36	360 ± 30
PA11/B2/PA11	1300 ± 54	280 ± 15

**Table 6 ijms-24-16737-t006:** Characteristics of the used materials.

Material	Melt Temperature (°C) *	Glass Temperature (°C) *	Density (g/cm^3^)	Average Molecular Weight, Mw (g/mol)
PLA	168	59	1.24	100,000
PA11	188	45	1.05	25,000
Joncryl	-	54	1.08 **	6800

* The glass and melt temperatures were determined using differential scanning calorimetry (DSC) analysis at 10 °C/min. ** The specific gravity as obtained from a technical data sheet of BASF.

**Table 7 ijms-24-16737-t007:** Compositions and designations of the unmodified and the modified PLA, PA11, and the blends.

Material	Blend Designation	PLA (wt%)	PA11 (wt%)	Joncryl (wt%)
PLA/PA11	(100/0/0)	100	0	0
	(80/20/0)	80	20	0
	(0/100/0)	0	100	0
PLA_Jonc	(99.3/0/0.7)	99.3	0	0.7
PLA/PA11/Jonc B1	(80/20/0.7)	79.44	19.86	0.7
PA11_Jonc	(0/99.3/0.7)	0	99.3	0.7
PLA_Jonc/PA11 B2	(80_0.7/20)	79.3	20	0.7
PLA/PA11_Jonc B3	(80/20_0.7)	80	19.3	0.7
PA11_Jonc/PLA_Jonc B4	(80_0.35/20_0.35)	79.65	19.65	0.7

## Data Availability

Data are contained within the article.

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
