# Peer review of "Biosourced Multiphase Systems Based on Poly(Lactic Acid) and Polyamide 11 from Blends to Multi-Micro/Nanolayer Polymers Fabricated with Forced-Assembly Multilayer Coextrusion"

_ijms, 2023, doi:10.3390/ijms242316737_

Round 1

Reviewer 1 Report

Comments and Suggestions for Authors

The paper is on the biosourced multiphase system of a PLA and PA11 blend compatibilized with Joncryl. The article is well written and has some scientific significance. I thoroughly enjoyed reading the paper. The paper is good for publication.I would like to request the author consider one of my comments to improve the quality of the paper.
The author mentioned studying the control of interface/interphase in the blended structures. Was the interphase quantified using any technique in this study? If not, I would suggest the author use the term "interface" only throughout the paper.

Author Response

Thank you for your positive feedback on the paper. I'm glad you found it well-written and scientifically significant. I appreciate your suggestion regarding the term "interface" and "interphase." While the study focused on the control of interface/interphase in the blended structures, the quantification of the interphase was not explicitly addressed in this study. I will consider your recommendation and use the term "interface" consistently throughout the paper to maintain clarity and precision.

Reviewer 2 Report

Comments and Suggestions for Authors

The reviewer read the manuscript entitled "Biosourced multiphase systems based on poly(lactic acid) and polyamide 11 from blends to multi-micro/nanolayer polymers fabricated by forced-assembly multilayer coextrusion". 

The contents were not bad, but should improve and revise the manuscript.

1. All Figures are low quaility. Please check the size and resolution.

2. Figure 8 and 9, the scale bar is significantly  important due to discussion about the size of structure. However, the reviewer could not find the scale bar, therefore, it is impossible to discuss the phase structure PLA/PA.

3. The author mentioned "strain hardening" from Figure 6. The authors should discuss about the condition of "strain hardening" Why Figure 6h has no "strain hardening"?

Author Response

The reviewer read the manuscript entitled "Biosourced multiphase systems based on poly(lactic acid) and polyamide 11 from blends to multi-micro/nanolayer polymers fabricated by forced-assembly multilayer coextrusion". The contents were not bad, but should improve and revise the manuscript..

Point 1: All Figures are low quaility. Please check the size and resolution.

Response 1: thanks for your comment, we apologize for any inconvenience. We have carefully reviewed the size and resolution of the figures and ensure that higher quality versions are provided in the final version of the manuscript.

Point 2: Figure 8 and 9, the scale bar is significantly  important due to discussion about the size of structure. However, the reviewer could not find the scale bar, therefore, it is impossible to discuss the phase structure PLA/PA.

Response 2: Thanks for bringing this to our attention, we have carefully added scale bars to these figures in the revised version to ensure clarity and accuracy in our analysis of the phase structure.

Point 3: The author mentioned "strain hardening" from Figure 6. The authors should discuss about the condition of "strain hardening" Why Figure 6h has no "strain hardening"?.

Response 3: Thank you for pointing out this aspect for further clarification, for the blend PLA/PA11-JCL, the compatibility between PLA and modified PA11-Joncryl can affect their ability to form a
